# Cuticular Hydrocarbon Plasticity in Three Rice Planthopper Species

**DOI:** 10.3390/ijms22147733

**Published:** 2021-07-20

**Authors:** Dan-Ting Li, Xiao-Jin Pei, Yu-Xuan Ye, Xin-Qiu Wang, Zhe-Chao Wang, Nan Chen, Tong-Xian Liu, Yong-Liang Fan, Chuan-Xi Zhang

**Affiliations:** 1Institute of Insect Science, Zhejiang University, Hangzhou 310058, China; 11516079@zju.edu.cn (D.-T.L.); yeyuxuan@zju.edu.cn (Y.-X.Y.); 11917011@zju.edu.cn (X.-Q.W.); zhechaowang@zju.edu.cn (Z.-C.W.); 2State Key Laboratory for Managing Biotic and Chemical Threats to the Quality and Safety of Agro-Products, Institute of Plant Virology, Ningbo University, Ningbo 315211, China; 3State Key Laboratory of Crop Stress Biology for Arid Areas and Key Laboratory of Integrated Pest Management on Crops in Northwestern Loess Plateau, Ministry of Agriculture, Northwest AandF University, Yangling 712100, China; xiaojinpei@nwafu.edu.cn (X.-J.P.); txliu@nwafu.edu.cn (T.-X.L.); 4Guangdong Provincial Key Laboratory of Insect Developmental Biology and Applied Technology, Institute of Insect Science and Technology, School of Life Sciences, South China Normal University, Guangzhou 510631, China; nanchen@m.scnu.edu.cn

**Keywords:** rice planthopper, cuticular hydrocarbons, transcriptome, temperature, humidity, host plant

## Abstract

Insect cuticular hydrocarbons (CHCs) are organic compounds of the surface lipid layer, which function as a barrier against water loss and xenobiotic penetration, while also serving as chemical signals. Plasticity of CHC profiles can vary depending upon numerous biological and environmental factors. Here, we investigated potential sources of variation in CHC profiles of *Nilaparvata lugens*, *Laodelphax striatellus* and *Sogatella furcifera,* which are considered to be the most important rice pests in Asia. CHC profiles were quantified by GC/MS, and factors associated with variations were explored by conducting principal component analysis (PCA). Transcriptomes were further compared under different environmental conditions. The results demonstrated that CHC profiles differ among three species and change with different developmental stages, sexes, temperature, humidity and host plants. Genes involved in cuticular lipid biosynthesis pathways are modulated, which might explain why CHC profiles vary among species under different environments. Our study illustrates some biological and ecological variations in modifying CHC profiles, and the underlying molecular regulation mechanisms of the planthoppers in coping with changes of environmental conditions, which is of great importance for identifying potential vulnerabilities relating to pest ecology and developing novel pest management strategies.

## 1. Introduction

Insect cuticular hydrocarbons (CHCs), which are composed of long, straight-chained, olefinic and methyl-branched hydrocarbons, act as a waxy coat that can adapt to environmental changes to prevent water loss and avoid the damage caused by chemical compounds (such as insecticides) [1,2,3]. CHCs are used as sex pheromones, kairomones, primer pheromones, and colony-, caste-, species- and sex-recognition signals in a wide range of insects [4]. Due to their widespread importance in waterproofing and chemical communication, CHCs are indispensable for insects and shaped by two different selection pressures [5]. Firstly, CHCs are affected by waterproofing requirements of insect habitats [6,7]. Secondly, CHCs are shaped by selection of their communication function from intraspecific and interspecific interactions [8,9]. Thus, CHCs provide classic examples of ‘magic’ and ‘dual’ traits that affect both ecological divergence and mating signals, and could lead to reproductive isolation and the formation of new species [10,11]. However, existing research has mostly been limited to ants [5,7,12], wasps and bees [13,14,15], and fruit flies and blowflies [11,16]. It is evident that the CHC profiles vary greatly between different insect groups. In certain ants and flies, CHC diversification between related species occurs gradually, typically resulting in greater similarity in substances and concentrations between closely related species compared to distant relatives [17,18]. Conversely, in some ants, wasps and stick insects, CHC diversification can also occur saltationally, where rapid changes between closely related species result in drastic differences in their CHCs and concentrations [5,14,19].

CHCs provide enormous functional significance in insects which have evolved a wide range of CHCs differing in chain length, methyl branching pattern, and position and number of double bonds [20,21]. Gibbs (1998) developed a theory for the assessment of the relationship between HC variation and its water retention ability, and the HCs with higher melting temperature (*T_m_*) may be more adept in waterproofing, especially at higher temperatures. In general, HCs with longer chain lengths, higher saturation, and fewer methyl branches present high *T_m_* and strong waterproofing ability [10,22]. CHC profiles can vary depending upon numerous environmental (i.e., temperature, humidity, diet) and biological factors (i.e., the developmental stage, sex, geographic origin of species or populations, and genetic background) [20,23,24]. Insects can prevent critical water loss in response to high temperature and low humidity by a fast change of their CHC phenotype [25]. Stinziano et al. (2015) found that rapid desiccation hardening treatment could reduce cuticular water loss via an acute change in cuticular hydrocarbons that enhance desiccation tolerance in *Drosophila melanogaster* [26]. In the same year, Dembeck et al. (2015) concluded that RNAi knockdown of 24 genes associated with fatty acid metabolism led to CHC variations in the flies. Moreover, these genes could be pieced together to make and recycle CHCs in a network [27].

The insect cuticular lipid biochemical pathway, as it is known, involves an enzymatic cascade including acetyl-CoA carboxylase (ACC), fatty acid synthase (FAS), fatty acid desaturase (FAD), fatty acid elongase (ELO), Fatty acyl-CoA reductase (FAR), P450 oxidative decarbonylase (CYP4G) and wax synthase (WS) [10]. RNAi knockdown against the unique ACC in *Drosophila*, CHCs were almost fully depleted in the flies [28]. RNAi knockdown of the microsomal FAS (CG3524) in the oenocytes also eliminates the production of methyl-branched CHCs in *D. melanogaster* [10]. In addition to this, RNAi knockdown of *BgFas1* causes a dramatic reduction of methyl-branched CHCs and a slight decrease of straight-chain CHCs [29]. Our previous studies demonstrated that knockdown of four ELO and two FAR genes resulted in a decrease of CHC amounts in *Nilaparvata lugens* [30,31]. Furthermore, reduction or elimination concerning the activity of the CYP4G encoded by *CYP4G76* and *CYP4G115* in *N. lugens*, *DmCYP4G1* in *D. melanogaster*, *LmCYP4G102* in *Locusta migratoria*, *CYP4G19* in *Blattella germanica* and *CYP4G51* in *Acyrthosiphon pisum* is responsible for the last step of CHC biosynthesis, and provokes a decrease in CHC amounts [32,33,34,35,36]. Nevertheless, the alterations in CHC composition as a consequence of gene disruption were often complex, causing some unexpected results. Apart from a few well documented case studies of unsaturated CHCs in *Drosophila*, we have not yet resolved individual genetic factors exclusively governing the synthesis of particular CHCs. The genetic basis of insect cuticular hydrocarbon biosynthesis and variation is still far from fully and conclusively understood.

The brown planthopper (*N. lugens*, BPH), small brown planthopper (*Laodelphax striatellus*, SBPH) and white-backed planthopper (*Sogatella furcifera*, WBPH) are three of the most destructive insect pests belonging to Delphacidae in Hemiptera. Although all of the three species take rice as their main food source, SBPH and WBPH have wide-ranging hosts, such as maize and wheat. At the same time, their tolerance to temperature differs significantly. In general, desiccation stress leads to an adaptive shift towards increased levels of longer chain CHCs, a higher proportion of saturated CHCs, and/or greater proportions of straight-versus branched-chain CHCs [20]. Moreover, the bodies of the insects are different in size either at different developmental stage or among the species, so the CHC amounts may vary greatly with their body size. As differences in CHC profiles of many pest species are primary cues of recognizing and potentially discriminating between sexes, developmental stages, related and unrelated individuals, populations, and species, determining the factors associated with CHC variation in insects is an important general step toward understanding their evolution and their potential utility in enhancing available pest monitoring and management strategies. Although there are many studies that concentrate on the effect of environmental factors on the chemical composition of insect cuticles, few studies have investigated the effect of environmental factors on the genes of HC biosynthetic pathways. The planthopper is one of the few insect species whose genes and enzymes involved in the HC biosynthetic pathway have been nearly completely understood by us. In this present study, by taking the opportunity of our understanding of the HC biosynthetic pathway, we investigated factors that might be associated with variation in CHC profiles in three planthopper species. To understand the underlying mechanisms, the transcriptomes of the three species were sequenced and compared under treatments of different environmental factors. Then, significant differences of biological and ecological effects in modifying biosynthetic pathway genes of CHCs in three planthopper species were found, which is conductive to revealing the molecular mechanisms shaping the CHC plasticity of plant hoppers in particular, and insects in general.

## 2. Results

### 2.1. Effects of Planthopper Species, Developmental Stages and Sexes on Cuticular Hydrocarbon Profiles

BPH, SBPH and WBPH were featured with similar qualitative hydrocarbon profiles. Extracts of three planthopper adult species contained 28 straight-chain alkanes that ranged from C_10_ to C_38_ in detectable quantities (Appendix A). The dominant hydrocarbons (comprising >1% of all *n*-alkanes) were *n*-cetane (C_16_), *n*-octodecane (C_18_), *n*-eicosane (C_20_), *n*-docosane (C_22_), *n*-pentacosane (C_25_), *n*-heptacosane (C_27_), *n*-octacosane (C_28_), *n*-nonacosane (C_29_) and *n*-hentriacontane (C_31_) (Appendix A). C_29_ was the main CHC component, i.e., accounted for 29.49%, 62.26% and 45.72% of all of the total *n*-alkanes in BPH, SBPH and WBPH female adults, respectively. Surprisingly, the main CHC components were C_16_ (22.33%) and C_18_ (23.07%) in BPH male adults, while the dominant component was C_29_ accounting for 53.87% and 40.46% in SBPH and WBPH male adults, respectively. For the fifth- and second-instar nymphs, the largest proportions of C_29_ were 39.43% and 39.99%, 35.67% and 33.49%, and 48.11% and 36.81% in BPH, SBPH and WBPH, respectively.

For the purpose of clarifying the effects of planthopper species, developmental stages and sexes on CHC profiles, the CHC qualities and quantities in female and male adults and the fifth-instar (3 d) and second-instar (2 d) nymphs were studied. Further, principal components (PCs) of variable *n*-alkanes within each species were computed to reduce the dimensionality of the data to orthogonal PCs. The first two principal components (PC1 (88.8%) and PC2 (5.2%)) explain 94% of the total variance in cuticular *n*-alkane composition among three species at different sexes and stages (Figure 1A). In addition, the first two PCs accounted for 89.55%, 98.21% and 95.25% of the total variation for BPH (Figure 1B), SBPH (Figure 1C) and WBPH (Figure 1D) cuticular *n*-alkanes, respectively. As for *n*-alkanes that contribute to principal component separation, these included different hydrocarbons, and the main component, C_29_, had contributed the most to PC1.

One-way ANOVA of the three planthopper species with different developmental stages revealed that there were significant differences among BPH, SBPH and WBPH of female and male adults and fifth- and second-instar nymphs for the total amount of cuticular *n*-alkanes per insect or milligram of fresh body mass (Figure 2). Comparing the total CHC components per insect between females and males of the same species, it was revealed that there was only a greater reduction in SBPH male adults than female adults, and no significant difference appeared in the BPH or WBPH (Figure 2A). However, when the total CHC components per milligram of fresh body mass were compared, it was found that the levels of male adults were higher than that of female adults among the three planthoppers (Figure 2B).

### 2.2. Effects of Temperature and Relative Humidity on Cuticular Hydrocarbon Profiles

The influence of different temperatures and relative humidity on cuticular hydrocarbon profiles of these three planthopper female adult species was investigated in detail. The first two PCs covered 84.74%, 97.28% and 95.71% of the total variation among BPH, SBPH and WBPH for female CHCs, respectively (Figure 3). In addition, effects of temperature and relative humidity on the amount of total cuticular *n*-alkanes and the main contributor (C_29_) were also analyzed by the Student’s *t*-test (Figure 4). For high temperature (38 °C) treatment, total cuticular *n*-alkanes and C_29_ amounts of SBPH adult females were significantly reduced when compared with the control (27 °C), while no significant difference between BPH or WBPH treatments was observed. For a low temperature (5 °C) treatment, the amount of total cuticular *n*-alkanes and C_29_ were significantly increased in WBPH, while no significant difference was detected between BPH or SBPH. Interestingly, the trends of the total amount of *n*-alkanes in high/low relative humidity were similar with those in high/low temperature treatments of all three planthoppers. The amounts of total *n*-alkanes and C_29_ were notably decreased in SBPH female adults under 98% RH, compared with the control (65% RH). Differently, the amounts in high/low relative humidity were significantly increased in WBPH, compared with the control (65% RH). In addition, the amount of C_29_ per insect in BPH also significantly decreased under 98% RH, while amounts of total *n*-alkanes and C_29_ significantly increased under 5% RH, compared with the control (65% RH).

### 2.3. Effects of Different Host Plants on Cuticular Hydrocarbon Profiles

As SBPH and WBPH are polyphagous insects, the CHC profiles of the female and male adults, fifth-instar and second-instar nymphs fed on rice or wheat crops, were investigated. The first two principal components (PC1 (88.1%) and PC2 (5.05%)) explain 93.15% of the total variance in cuticular *n*-alkane composition between the rice group and the wheat group (Figure 3D). The *n*-alkanes contributing to PC separation were the dominated cuticular components, i.e., C_29_, C_28_ and C_27_. Then, the amount of total cuticular n-alkanes and C_29_ per insect and milligram of fresh body mass were analyzed by Student’s *t*-test, respectively (Figure 5). In SBPH, data from GC/MS confirmed that there were significant differences in the amount of total cuticular *n*-alkanes and C_29_ per insect between rice- and wheat-SBPH at different developmental stages. Taking the weight into account, there were also significant differences of C_29_ amounts between rice- and wheat-SBPH at different developmental stages, while only a significant difference of total amounts of cuticular *n*-alkanes between rice- and wheat-SBPH at the fifth-instar stage. In WBPH, there were significant differences in the amount of total cuticular *n*-alkanes and C_29_ between rice- and wheat-WBPH female adults and second-instar nymphs according to both calculation methods.

### 2.4. Effects of Temperature and Relative Humidity on Differential Gene Expression in Three Planthopper Species

Using the Illumina sequencing platform NovaSeq 6000, three temperature and three relative humidity treatments were performed separately on the three planthopper species, which generated a total of 54 gene expression libraries (NCBI SRA: PRJNA693158), including three biological replicates of each treatment. The variations in gene expression were analyzed in comparisons of 5–27 °C, 27–38 °C, 5%–65% RH and 65%–98% RH in the three species. The details of down-regulation/up-regulation genes are described in File S1. For 5 °C treatment, a total of 2648, 134 and 473 genes were differentially expressed with an absolute value of the log2 ratio > 1 in BPH, SBPH and WBPH, respectively (Figure 6A). In BPH, 1570 genes were up-regulated and 1078 genes were down-regulated, respectively. Genes associated with cuticular proteins, fatty acid synthesis, fatty acid elongation and cytochrome P450 were highly up-regulated. Most of the down-regulated genes were related to transcription, such as zinc finger protein, transcriptional regulators and transcription factors. In SBPH, 100 genes were up-regulated and 34 genes were down-regulated, respectively. Up-regulated genes related to cuticular protein and cytochrome P450 were similar to those of BPH. Interestingly, genes related to cuticular protein and cytochrome P450 were down-regulated in WBPH. Up-regulated genes (197 genes) such as esterase, FAD and FAS were also differentially expressed between BPH and SBPH.

In comparisons between 38 °C and 27 °C, a total of 2373 genes showed significant differential expression in BPH, with 1531 genes up-regulated and 860 genes down-regulated (Figure 6A). Clearly, a large number of PiggyBac transposable element-derived protein genes were up-regulated, as well as the genes for glucose metabolism, cytochrome P450, cuticular proteins and heat shock proteins (HSPs). It is of note that CHC biosynthesis related genes, such as FAS and ELO genes, were either up-regulated or down-regulated at the high temperature, which might be the reason accounting for complementary functions of different genes in a same family. In SBPH, a total of 519 genes were significantly differentially expressed, with 214 genes up-regulated and 305 genes down-regulated. HSP genes in SBPH showed a similar expression pattern to that in BPH. Genes related to cuticular protein and cytochrome P450 were down-regulated, though they were up-regulated in the cold treatment. In addition, FAD and FAR genes were also significantly down-regulated. In WBPH, 757 genes were up-regulated. Among them, zinc finger proteins (i.e., zinc finger protein 12-like, zinc finger protein 23-like and zinc finger protein 26-like) and HSP genes (HSP68-like, HSP70 and HSP90) showed significant altered expressions. In the meantime, histone genes (histone H1-like, histone H2B and histone H4), which are necessary for the segregation of chromosomes during mitosis, were up-regulated under the high temperature, indicating an increased cell proliferation process. A total of 873 genes down-regulated under high temperature were detected in WBPH.

For relative humidity treatment experiments, a total of 610, 309 and 209 genes were differentially expressed in BPH, SBPH and WBPH, respectively (Figure 6B). In BPH, 550 genes were up-regulated while 60 genes were down-regulated. Among them, piggyBac transposable element-derived proteins showed a significant altered expression, which was similar to that of BPH in the high temperature treatment. In SBPH, 258 genes were up-regulated and 51 genes were down-regulated. Genes relevant to cuticular protein and HSP were highly up-regulated. Although unlikely, the number of up-regulated genes (86 genes) was less than that of down-regulated ones (123 genes) in WBPH. Under the high humidity condition, a total of 2251 genes displayed significant different expressions in BPH, when 1539 genes were up-regulated while 712 genes were down-regulated (Figure 6B). Like those in the relative humidity treatment, there were lots of PiggyBac transposable element-derived protein genes up-regulated. Genes associated with CHC synthesis, such as *FAS*, *ELO*, *FAR* and *cytochrome P450* were also significantly up-regulated. However, the expression of cuticle protein and histone genes was significantly down-regulated. In SBPH, only 134 genes showed significantly differential expressions, with 96 genes up-regulated and 38 genes down-regulated. Similarly, genes related to CHC synthesis were up-regulated, while those related to cuticular proteins and histones were down-regulated. A large number of cuticle protein genes were also down-regulated in WBPH. Differently, gene expression levels of histone and ELO were significantly repressed at a high relative humidity. Nevertheless, a lot of differentially expressed genes of these three planthoppers were featured with unknown functions, and they might also play important roles in response to different relative humidity treatments.

### 2.5. Different Host Plant Effects on Differential Gene Expression in SBPH and WBPH

In the comparisons of rice and wheat plants, a total of 403 and 466 genes were differentially expressed in SBPH and WBPH, respectively (Figure 7A). In SBPH, there were 325 up-regulated genes and 78 down-regulated genes. Obviously, genes encoding cuticle protein, larval cuticle protein, endocuticle structural glycoprotein, mucin, trypsin, cytochrome P450 (CYP417B1 and CYP426A1) and HSP (HSP68-like and HSP70-5) were up-regulated dramatically. However, histone genes (histone H1-like, histone H3-like and histone H2B) showed significantly lower expressions in SBPH fed on wheat plants. In WBPH, 238 genes were up-regulated and 228 genes were down-regulated, respectively. Among them, the expression levels of histone genes were also down-regulated, while the genes related to histone H1-Ⅱ-like, histone H2A-like, histone 4 and histone H4 were differentially expressed. Additionally, other down-regulated genes were related to cuticle proteins, carboxylesterase, HSP70 and some cytochromes P450 (CYP307A2, CYP404B1 and CYP439A3), while the up-regulated genes were related to CHC biosynthesis pathway genes, such as FAS, ELO, FAD, FAR and some cytochromes P450 (CYP6AX3 and CYP4C62).

### 2.6. GO Enrichment Analysis

To make the RNA-seq analysis on the CHC profiles more comprehensive, we searched the Gene Ontology (GO) functional terms for the differentially expressed genes of all three planthopper species compared among the treatments of different environmental factors. The GO annotations for molecular functions, cellular components and biological process categories are described in Appendix A. Then, the functional terms highly related to the cuticular lipid biosynthetic pathway from the current analysis were extracted (Figure 8). Under the low temperature, GO classification of BPH showed that most of the enriched gene sets from high to low levels belonged to the organic acid metabolic process > carboxylic acid metabolic process > lipid biosynthetic process > carboxylic acid biosynthetic process > organic acid biosynthetic process (Figure 8A). In WBPH, the top five highly enriched gene sets were the same as those in BPH. Different from the enriched gene sets in BPH and WBPH, only seven genes were enriched to the lipid biosynthetic process, carboxylic acid metabolic process, lipid catabolic process and organic acid metabolic process in SBPH. Under the high temperature, there were fewer enriched genes in BPH but much more in WBPH and SBPH than under the low temperature (Figure 8B).

For high relative humidity, the results of GO enrichments were similar to those of BPH and WBPH under the high temperature, while no genes were enriched to the cuticular lipid biosynthetic pathway gene set in SBPH (Figure 8C). For the low relative humidity, there were fewer genes correlated with cuticular lipid biosynthesis enrichment (Figure 8D).

GO enrichment analysis on the differentially expressed genes between rice- and wheat-feeding SBPH and WBPH, respectively, demonstrated that the number of genes enriched to the cuticular lipid biosynthetic pathway gene set in WBPH was larger than that of SBPH (Figure 7B). Nevertheless, the largest enriched gene sets in these two groups were the same with the organic acid metabolic process > carboxylic acid metabolic process > carboxylic acid biosynthetic process > organic acid biosynthetic process.

### 2.7. Cuticular Lipid Biosynthetic Pathway Gene Expression Profiles in the Three Species

Heatmaps representing the expression profiles of cuticular lipid biosynthetic pathway genes in female adults of BPH, SBPH and WBPH were constructed, and at the same time they also clearly showed differences between the treatments of different environmental factors. From our previous study, four ELO (*NlELO2*, *3*, *8* and *16*) and two FAR (*NlFAR7* and *9*) genes essential for cuticle waterproofing in BPH were discovered. GC/MS quantification indicated that knockdown of each of these six genes resulted in a decrease of the CHC amount [30,31,36]. In BPH, *NlELO2*, *NlELO16* and *NlFAR7* were up-regulated with various gene expression levels under different environmental conditions, while *NlELO3*, *NlELO8* and *NlFAR9* were down-regulated (Figure 9). Unlike what was mentioned before, *LsELO2*, *LsELO16*, *LsFAR7* and *LsELO3* were down-regulated at 5 °C and 38 °C, or under 98% RH in SBPH, while *LsELO8* and *LsFAR9* were up-regulated. In WBPH, *SfFAR7* was down-regulated under the above different conditions, while *SfFAR9* was up-regulated. There was no significant difference in the expression level of *SfELO2*, *SfELO3* and *SfELO16* except for the down-regulation of *SfELO3* and *SfELO16* at 38 °C, and *SfELO2* at 5 °C.

## 3. Discussion

Since CHCs represent the boundary between the organism and its environment, and cuticular lipid components can change while offering protection against adverse environmental conditions, variation in CHC profiles may present a target for natural selection and adaptive evolution. Here, causes of variation in CHCs in three planthopper species were explored, and considerable phenotypic CHC variation attributed to variations of developmental stages, sexes, temperature, humidity and host plants was also detected. Then, by performing RNA-seq analysis, the differential gene expression of the three planthoppers under different environmental conditions was compared. Therefore, in this study, a comprehensive and accurate gene expression profile for CHC variation is presented by different environmental and biological factors in three very important agricultural pest insects.

### 3.1. Exploring Factors Associated with Variation in CHC Profiles among the Planthopper Species

CHCs, existing in almost all insects, are highly diverse and remarkably species-specific [12]. Studies in the literature confirm that CHCs consisting of straight-chain, methyl-branched, and unsaturated hydrocarbons are the main components of the insect surface lipid layer [37,38]. Based on our findings, it can be seen that straight-chain hydrocarbons (C_10-38_) had the highest number and percentage of compounds among the three planthopper species. Duarte et al. (2019) reported that linear alkanes with longer chains may be linked to waterproofing to protect ant species from desiccation [39]. Our previous studies also indicated that linear alkanes were the vast majority of CHCs in BPH, and CHC deficiency resulted in increased adhesion of water droplets to the animal surface and the inability to survive in paddy fields [30,31]. Therefore, in the present study, the analysis on CHC profiles among the different planthopper species was focused on the *n*-alkanes. Although the types of CHCs among three planthoppers are similar, the percentage of each alkane was different with a varying degree. PC analysis on *n*-alkanes within each species and among different species showed that CHC profiles can vary with developmental stages, sexes and insect species. Generally, the weight of the insect increases with the body size (female adults and fifth-instar nymphs > male adults > second-instar nymphs), and the body size of BPH is bigger than that of WBPH than SBPH. As expected, the level of total alkanes increased with the insect growth, except for the peak value occurring at the fifth-instar stage of BPH. We found sex-related differences between female and male adults, as the amount of CHCs per milligram of fresh body mass in male adults significantly increased among three species. The dominant CHC component C_29_ was changed with C_16_ and C_18_ in BPH male adults, while no significant change was observed in SBPH and WBPH. As there has been no report yet to indicate the CHC function as sex pheromones in Hemipteran insects, whether these changes between female and male adults are closely correlated with courtship behavior needs further research in the future.

After this, the CHC profiles of wheat-feeding SBPH and WBPH at different developmental stages and sexes were investigated. PCA exhibited that CHC profiles in both wheat-feeding SBPH and WBPH were notably different from those of rice-feeding insects, respectively. Furthermore, the mean values of total alkanes and the most dominant C_29_ per insect in wheat-feeding SBPH and WBPH were significantly increased when compared with the rice-feeding SBPH at the second-instar stage. However, the effects were various in these two planthoppers with significantly low levels in wheat-feeding SBPH at fifth-instar and adult stages, and high levels in wheat-feeding WBPH at female adult stages. Until the present, numerous studies have demonstrated that the type of diet played a significant role in CHC variations in many insect species. For example, when the mustard leaf beetle *Phaedon cochleariae* switches to a novel host plant species, its novel CHC profile differs from the former after two weeks to a great extent [40]. A switch between two distinct environments (e.g., host plant switch by phytophagous insects) causes stepwise formation of two distinct adaptive phenotypes, while a gradual environmental change (e.g., temperature gradients) induces a gradual change of numerous adaptive CHC phenotypes [20].

Subsequently, temperature and humidity were documented as the environmental factors associated with variations in CHC profiles in the planthopper species, as has been shown previously in numerous other insect orders, including *D. melanogaster* and ant species (*Atta sexdens*, *Odontomachus bauri* and *Ectatomma brunneum*) [26,39]. When temperatures rose above 35 °C, *A. sexdens* workers began to die, and the production of linear alkanes increased in an attempt to satisfy the need for waterproofing [39]. Huang et al. (2017) had tested the survivorship of BPH, SBPH and WBPH, and then found that all of the species survived for a much shorter duration at 37 °C than at 27 °C, and 77.5% of SBPH survived for over 40 days, while all WBPH or BPH died within 15 days under 5 °C [41]. Our results showed that there existed significant differences in CHC profiles of planthoppers among different temperature and humidity treatments, as indicated by PCA. Furthermore, the mean values of total alkanes and C_29_ per insect or milligram of fresh body mass showed that there were significant reductions only in SBPH at 38 °C and 98% RH, and significant increases in WBPH at 5 °C and 5% RH, compared with the control. In this case, it was suggested that the influence of high temperature/humidity on CHC variation in SBPH was more pronounced, but was converse in WBPH, which may be due to the inhabiting and overwintering geographical areas of the three species which are quite different. The northern borders of overwintering areas for BPH and WBPH are around 21–25 °N, but SBPH can overwinter in all rice-growing areas in Asia [41]. In addition, the amounts of total alkanes and C_29_ per milligram of fresh body mass were also significantly increased in BPH and SBPH under 5% RH, while there was no significant difference in the amounts calculated by the number of insects, which may result from weight reduction of insects under desiccation. In general, desiccation stress can lead to an adaptive shift towards increased levels of longer chain CHCs and a higher proportion of saturated CHCs in insects [10,20].

### 3.2. Searching for Clues Linking CHC Variation to Different Environmental Factors in the RNA-Seq Analysis

Despite the rich diversity of CHC in insects, the main biosynthetic pathway for all CHCs is conserved [37]. In most organisms, the CHC synthesis pathway is presumably co-opted from the fatty acid synthesis pathway, and consecutively associated with ACC, FAS, ELO, FAD, FAR, CYP4G and WS gene families [10]. As all of the CHCs share a common biochemical pathway, differential expressions of the genes in this pathway could lead to the diversification of CHC profiles between insect species. According to our previous identification of cuticular lipid biosynthesis pathway genes in BPH, 53, 40 and 36 genes correlated with CHC synthesis in BPH, SBPH and WBPH were identified, respectively. In this study, the transcriptional responses to the heat and cold and humid and desiccant environments in three planthopper species were also compared. Several genes correlated with CHC synthesis were differentially expressed when exposed to the low and high temperature/humidity. Subsequently, it was found that *NlFAS2*, *NlFAD3*, *NlELO2* and *NlELO16* were significantly up-regulated under both low and high temperatures in BPH, and the up-regulated genes in WBPH were related to ACC, FAS and FAD, but in SBPH they were just related to *LsELO18* under 38 °C. Besides, it was also revealed that *NlFAR9*, *NlFAD2*, *4* and *NlELO4*, *5*, *7* were down-regulated under the low temperature, while no cuticular synthesis pathway genes were significantly differentially expressed in SBPH and WBPH. Furthermore, it is important to note that the down-regulated genes (*LsACC*, *LsFAS2* and *LsFAD1*) in SBPH were different from those in BPH and WBPH. In combination with the reduction of CHCs under the high temperature in SBPH, it was hypothesized that these three genes and the up-regulated *LsELO18* may play a key role in CHC synthesis in SBPH. In addition, *LsELO18* and *LsELO5* were also up-regulated at 98% RH in SBPH, while the total amount of CHCs was significantly decreased. In WBPH with the increased amount of CHCs under 98% RH, the expression levels of *SfACC* and *SfFAS2* were increased dramatically, while no other cuticular synthesis pathway genes were significantly expressed. Consequently, these two genes appear to be associated with the synthesis of CHCs in WBPH under high relative humidity. In BPH, the content of C_29_ was decreased to a great extent as well as the expression level of *NlFAD4* under the high relative humidity, suggesting that *NlFAD4* may be important for producing C_29_. Under desiccation, the amount of CHCs was significantly increased in three planthoppers, while only *NlFAS2* and *NlELO2* in BPH, *SfACC* and *SfFAD* in WBPH, and none in SBPH were notably up-regulated. These results indicate that cuticular lipid biosynthesis pathway genes and other lipid metabolic process genes could be pieced together in a network to thicken the surface layer. Although there were many cuticular lipid biosynthesis pathway genes significantly differentially expressed in the treatments, the amount of total CHCs had no significant change. One reason may be the complementary functions of the cuticular lipid biosynthesis pathway genes, while the other reason may be that it takes much more time for the change of CHC amounts to switch the environments in these three insects. Furthermore, there were no differential expression genes that were found between wheat- and rice-feeding SBPH, while the amounts of total *n*-alkanes and C_29_ were significantly decreased in wheat-feeding SBPH. Conversely, the amounts of *n*-alkanes and C_29_ were largely increased in wheat-feeding WBPH female adults, and *SfACC*, *SfFAS2* and *SfELO5* were dramatically up-regulated. As the expression level of *SfACC* and *SfFAS2* was increased in WBPH under a low/high temperature and humidity without significant change of CHC amounts, it was hypothesized that *SfELO5* was essential for the production of CHC in WBPH.

Moreover, genes, including those related to cytochromes P450, cuticular protein, histone and HSP, were found which showed differential expression in different environments, suggesting that their functions were associated with the adaptation of planthoppers to the environment. Huang et al. (2017) had reported that genes related to metabolism, transportation, exoskeleton and chemosensing are modulated when compared to the transcriptomes of the newly emerged females in three planthopper species treated under different temperatures for 24 h [41]. Though the age of adults being used for RNA-seq is different, the responses and tolerance mechanisms of these three planthopper species in coping with environmental change still have something in common. However, a large number of differentially expressed genes of these three planthopper species showed unknown functions, which might also exert key roles in temperature treatment. The mechanism on how the modulated genes cause CHC variations remains to be further investigated.

## 4. Materials and Methods

### 4.1. Maintenance of Planthopper Strains

The BPH, WBPH and SBPH used in this study were originally obtained from rice fields in Hangzhou (30°16′ N, 120°11′ E), China. As they are migratory pests, it was difficult to distinguish the geographical sources of the planthoppers. To weaken the influence of geographical factors, the samples used in this experiment were the offspring from a single female that was reared at 26 ± 1 °C and 60 ± 5% relative humidity on rice (Xiushui 134) and wheat (Fengdecun 1) seedlings under a 16:8 h (light:dark) photoperiod for more than 20 generations.

### 4.2. Set-Up of Thermal and Humid Experiments

For the thermal experiments, three groups of 150 newly emerged females each were reared on 3 leaf-stage rice seedlings at 27 °C and 65% relative humidity under a 16-h:8-h light:dark cycle for 2 days, and then two of the groups were transferred to 5 °C and 38 °C with 65% relative humidity under a 16-h:8-h light:dark cycle for 24 h, respectively. Samples were collected from a set of 15 insects to evaluate the CHCs amounts and another 10 insects to analyze transcriptomes. Three independent replications were performed for each treatment of three species.

For the humidity-treated experiments, three groups of 150 newly emerged females each were reared on 3 leaf-stage rice seedlings at 27 °C and 65% relative humidity under a 16-h:8-h light:dark cycle for 2 days, and then two of the groups were transferred to 5% and 98% relative humidity at 27 °C under a photoperiod of 16:8 h (light:dark) for 24 h, respectively. Samples were collected from a set of 15 insects to evaluate the CHCs amounts and another 10 insects to analyze transcriptomes. Three independent replications were performed for each treatment of three species.

### 4.3. Extraction and Quantification of CHCs

To investigate the CHC types and amounts of the planthopper species of different developmental stages on different hosts, CHCs were extracted from insects following a procedure described in our previous study [42]. Briefly, 60 s-instar nymphs, 15 fifth-instar nymphs, 15 female adults or 20 male adults (approximately 15 mg each) were immersed in 200 µL *n*-hexane in which 500 ng *n*-heneicosane (C_21_) was added as an internal standard. The solvent was stirred gently for 3 min and then drawn with a glass Pasteur pipette into a clean chromatography vial. This procedure was repeated twice, and finally used 200 µL hexane to rinse the nymphs and vial. All of the hexane extracts were combined, dried to 200 µL under high-purity nitrogen gas, and loaded onto a ~300 mg silica gel (70e230 mesh, Sigma-Aldrich, Louis, MO, USA) mini-column in a glass wool-stoppered Pasteur pipette. The HC fraction was eluted with 2 mL hexane, taken to dry absolutely under nitrogen gas, and resuspended in 100 µL hexane. The samples were analyzed on a TRACE 1310 (Thermo Scientific, Waltham, MA, USA) gas chromatograph (GC) equipped with an ISQ single quadruple MS and interfaced with Xcalibur 2.2 software.

### 4.4. Statistical and Quantitative Analyses

We counted the number and weight of insects in each sample. The quantities of CHCs were calculated according to the internal standard by two ways. One way is to calculate the mean value of CHC content of one insect (ng per insect), another is to calculate the mean value of CHC content of one sample by fresh insect body mass (ng/mg). Results from the comparative analyses were subjected to statistical analysis. Principal component analyses (PCA) were used to visualize overall treatment effects on hydrocarbon profiles, and treatment effects on the compounds that represented the top principal component was tested with Student’s *t*-test and one-way analyses of variance (ANOVA) in GraphPad Prism 8. Statistical differences were considered significant if *p* < 0.05. PCA was conducted in JMP Pro13 (SAS Institute Inc.).

### 4.5. cDNA Library Preparation and Illumina Sequencing

Total RNA was extracted using the TRIzol Total RNA Isolation Kit (Takara, Kyoto, Japan) according to the manufacturer’s protocol. RNA integrity was assessed using the RNA Nano 6000 Assay Kit of the Bioanalyzer 2100 system (Agilent Technologies, Santa Clara, CA, USA). A total amount of 1 μg RNA per sample was used as input material for the RNA sample preparations. Briefly, mRNA was purified from total RNA using poly-T oligo-attached magnetic beads. Fragmentation was carried out using divalent cations under elevated temperature in First Strand Synthesis Reaction Buffer (5X). First strand cDNA was synthesized using random hexamer primer and M-MuLV Reverse Transcriptase (RNase H). Second strand cDNA synthesis was subsequently performed using DNA Polymerase I and RNase H. Remaining overhangs were converted into blunt ends via exonuclease/polymerase activities. After adenylation of 3′ ends of DNA fragments, an adaptor with a hairpin loop structure was ligated to prepare for hybridization. In order to select cDNA fragments of preferentially 370–420 bp in length, the library fragments were purified with AMPure XP system (Beckman Coulter, Beverly, MA, USA). Then, a PCR was performed with Phusion High-Fidelity DNA polymerase, Universal PCR primers and Index (X) Primer. At last, PCR products were purified (AMPure XP system) and library quality was assessed on the Agilent Bioanalyzer 2100 system.

The clustering of the index-coded samples was performed on a cBot Cluster Generation System using a TruSeq PE Cluster Kit v3-cBot-HS (Illumia), according to the manufacturer’s instructions. After cluster generation, the library preparations were sequenced on an Illumina Novaseq platform, and 150 bp paired-end reads were generated.

### 4.6. Transcriptome and Differential Expression Analysis

The index of the reference genome was built using Hisat2 v2.0.5, and paired-end clean reads were aligned to the reference genome using Hisat2 v2.0.5. The mapped reads of each sample were assembled by StringTie (v1.3.3b) in a reference-based approach. StringTie uses a novel network flow algorithm as well as an optional de novo assembly step to assemble and quantitate full length transcripts representing multiple splice variants for each gene locus. TPM expression values were calculated using featureCounts for genes. Differential expression analysis was carried out using the DESeq2 package. The differentially expressed gene was judged for meeting the following conditions: false discovery rate (FDR) < 0.05 and absolute value of the log2 ratio > 1.

### 4.7. GO Enrichment Analysis of Differentially Expressed Genes

Gene Ontology (GO) enrichment analysis of differentially expressed genes was implemented by the clusterProfiler R package, in which gene length bias was corrected. GO terms with corrected *p*-value less than 0.05 were considered significantly enriched by differentially expressed genes.

### 4.8. Expression Profile Analysis of Cuticular Lipid Biosynthetic Pathway Genes

Based on a bioinformatics analysis, cuticular lipid biosynthetic pathway genes encoding ACC, FAS, ELO, FAD, FAR and CYP4G proteins were screened from the BPH, SBPH and WBPH genomic and transcriptomic databases by using the amino acid sequences of the reported genes from other insects, such as *D. melanogaster*, *Bombyx mori*, *Apis mellifera*, *Tribolium castaneum*, *Anopheles gambiae* and *Acyrthosiphon pisum* which were obtained from FlyBase, SilkDB and NCBI. Then, TPM expression values were used to analyze the expression profile of the cuticular lipid biosynthetic pathway genes by Heatmaps.

## 5. Conclusions

To conclude, results from our study show that CHC profiles in the three planthopper species vary depending on either developmental stages and sexes, or temperatures, humidity and host plants. From comparative analysis of the transcriptomes, it can be observed that the identification of candidate genes is associated with variation in CHC compositions and amounts in planthoppers treated with different environmental factors, including several genes which could be plausibly associated with CHC biosynthesis. Knowledge of these CHC profile variations and their underlying molecular regulation mechanisms may have potential applications in enhancing existing monitoring and management strategies of these important rice pests.

## Figures and Tables

**Figure 1 ijms-22-07733-f001:**
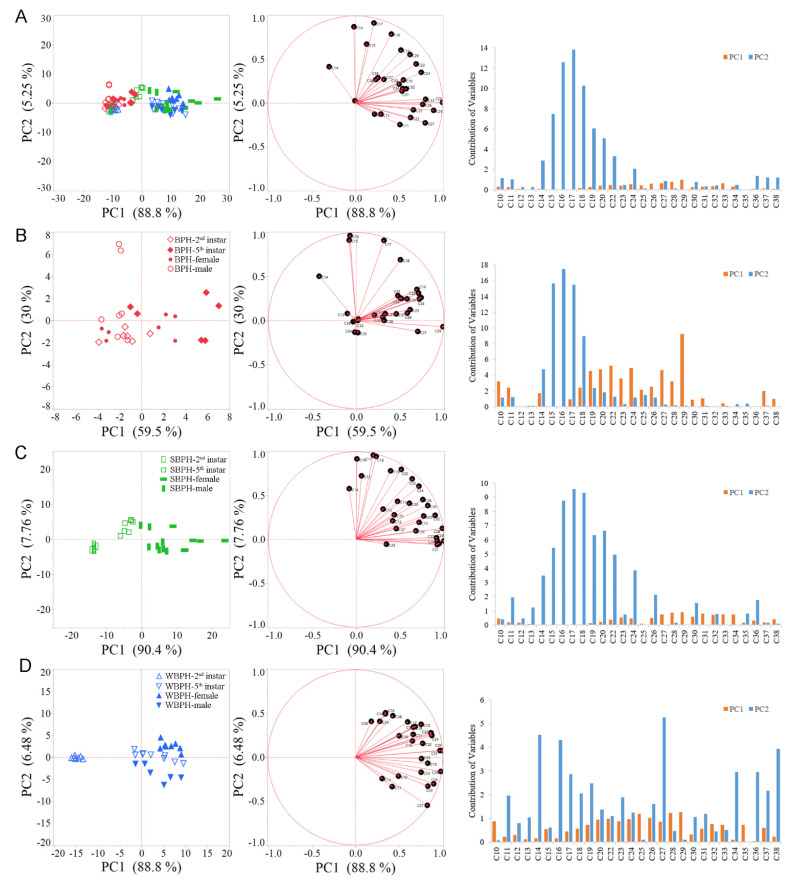
Principal component (PC) biplots for PC1 and PC2 of CHCs in the three planthopper species. (**A**) second- and fifth-instar nymphs and female and male adults of *Nilaparvata lugens* (BPH), *Laodelphax striatellus* (SBPH) and *Sogatella furcifera* (WBPH). (**B**) second- and fifth-instar nymphs and female and male adults of BPH. (**C**) second- and fifth-instar nymphs and female and male adults of SBPH. (**D**) second- and fifth-instar nymphs and female and male adults of WBPH. Left: The PCA shows PC1 and PC2 with the explained variance in brackets. Different groups are color-coded. Mid: eigenvectors of PC1 and PC2. Right: contribution of n-alkanes to PC1 and PC2 separation.

**Figure 2 ijms-22-07733-f002:**
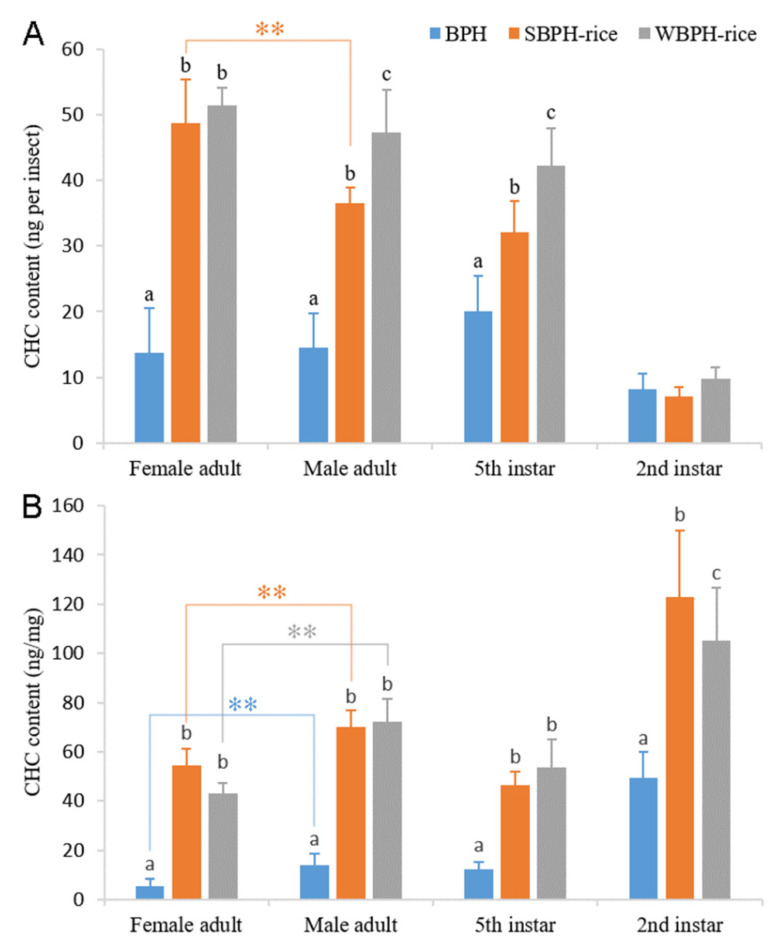
CHC contents of BPH, SBPH and WBPH at second- and fifth-instar and female and male adult stages. The contents of total straight-chain alkanes were analyzed by GC/MS. The results were calculated from ten biological replicates ((**A**): nanograms per insect ± SE, (**B**): nanograms per milligram of fresh body mass ± SE). Different letters above the error bars indicate significant difference among the stage group (ANOVA, LSD, *p* < 0.05). ** *p* < 0.01 (Student’s *t*-test).

**Figure 3 ijms-22-07733-f003:**
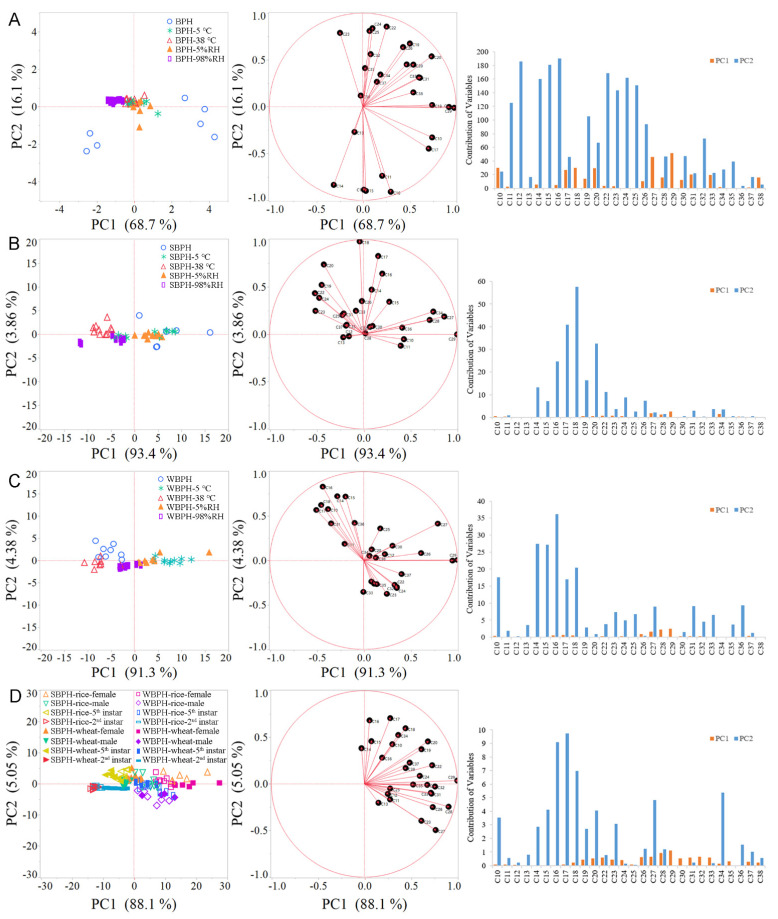
Principal component (PC) biplots for PC1 and PC2 of CHCs in the three planthopper species under different environmental conditions. (**A**) *Nilaparvata lugens* (BPH) in low/high temperature and humidity. (**B**) *Laodelphax striatellus* (SBPH) in low/high temperature and humidity. (**C**) *Sogatella furcifera* (WBPH) in low/high temperature and humidity. (**D**) wheat- and rice-feeding SBPH and WBPH at different stages. Left: the PCA shows PC1 and PC2 with the explained variance in brackets. Different environmental groups are color-coded. Mid: eigenvectors of PC1 and PC2. Right: contribution of n-alkanes to PC1 and PC2 separation.

**Figure 4 ijms-22-07733-f004:**
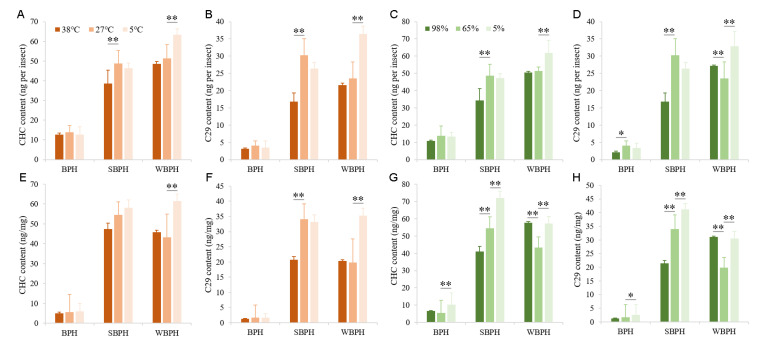
CHC contents of BPH, SBPH and WBPH treated with different temperatures (**A**,**B**,**E**,**F**) and humidity (**C**,**D**,**G**,**H**). The contents of total straight-chain alkanes and C_29_ were analyzed by GC/MS. The results were calculated from ten biological replicates (nanograms per insect or milligram of fresh body mass ± SE). * *p* < 0.05, ** *p* < 0.01 (Student’s *t*-test).

**Figure 5 ijms-22-07733-f005:**
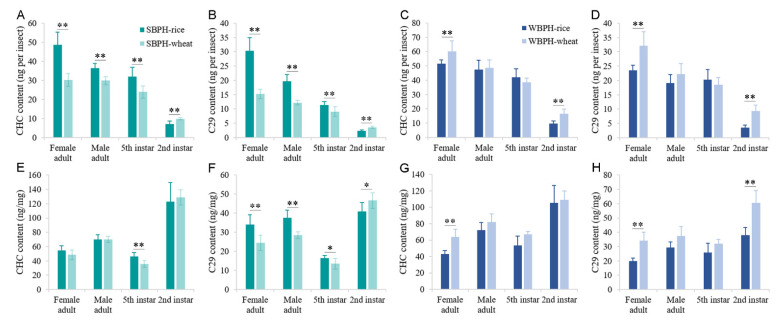
CHC contents of wheat- and rice-feeding SBPH (**A**,**B**,**E**,**F**) and WBPH (**C**,**D**,**G**,**H**) at different stages. The contents of total straight-chain alkanes and C_29_ were analyzed by GC/MS. The results were calculated from ten biological replicates (nanograms per insect or milligram of fresh body mass ± SE). * *p* < 0.05, ** *p* < 0.01 (Student’s *t*-test).

**Figure 6 ijms-22-07733-f006:**
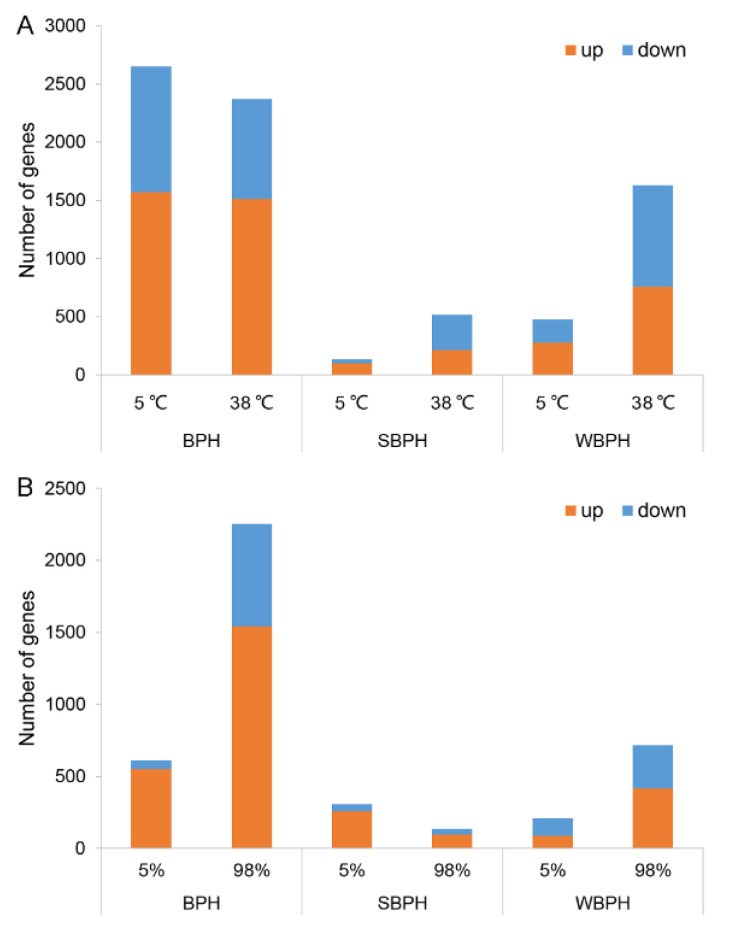
Number of differentially expressed genes in three planthoppers under 5 °C or 38 °C (**A**) and 5% or 98% relative humidity (**B**) compared with that at 27 °C with 65% RH, respectively. Up- (orange) and down-regulated (blue) unigenes were quantified.

**Figure 7 ijms-22-07733-f007:**
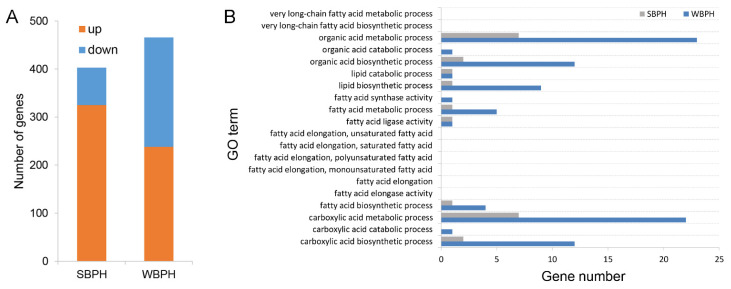
Comparative analysis of the transcriptomes between wheat- and rice-feeding SBPH and WBPH. (**A**) Number of differentially expressed genes in planthoppers feeding on different plants. (**B**) GO functional terms of differentially expressed genes in planthoppers feeding on different plants attributed to CHC biosynthetic pathways.

**Figure 8 ijms-22-07733-f008:**
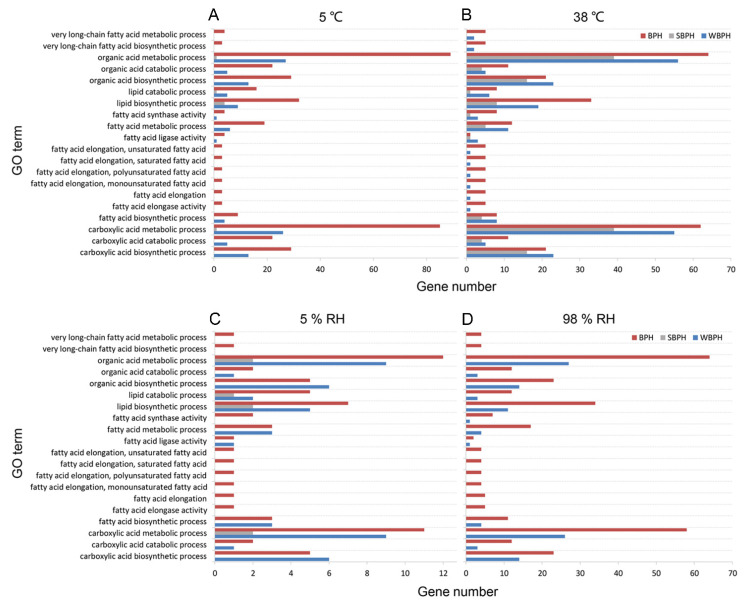
GO functional terms of differentially expressed genes in planthoppers under low/high temperatures (**A**) and relative humidity (**B**) attributed to CHC biosynthetic pathways.

**Figure 9 ijms-22-07733-f009:**
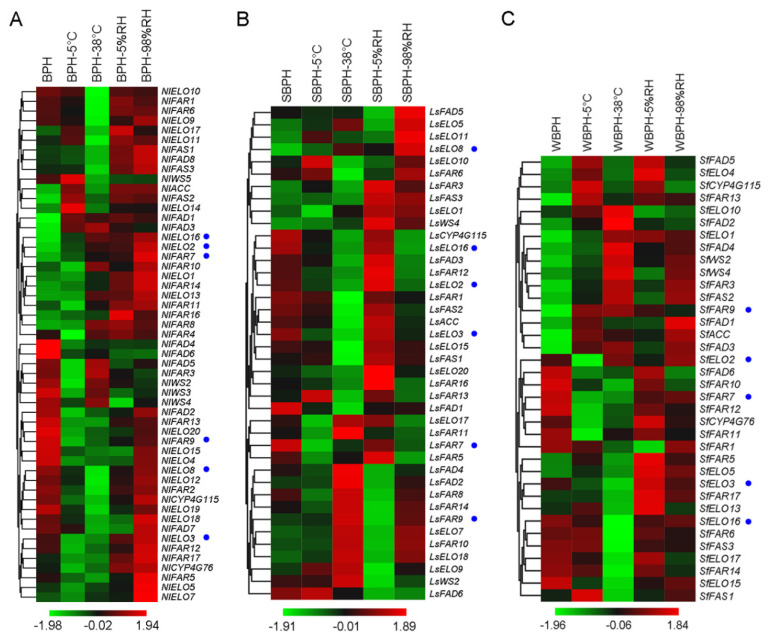
The expression profiles of cuticular lipid biosynthetic pathway genes in BPH (**A**), SBPH (**B**) and WBPH (**C**) under different treatments. The previous verified functional genes were indicated by blue dots.

## Data Availability

RNA-seq data are available in the National Center for Biotechnology Information (NCBI) under SRA accession number: PRJNA693158.

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
