# Peer review of "Cuticular Hydrocarbon Plasticity in Three Rice Planthopper Species"

_ijms, 2021, doi:10.3390/ijms22147733_

Round 1
Reviewer 1 Report
This is an ambitious manuscript aiming to cover a variety of topics and reflects a substantial amount of work by the authors. However, the results and M&M section are difficult to read for a non-specialist audience and may not provide enough information about the experimental design and analyses used. I strongly suggest revising the statistical analyses used in some sections and their interpretation. This paper has valuable information and would be suitable for publication after these issues have been addressed. You will find detailed comments in the attachment.

Reviewer 2 Report
The manuscript entitled ‘Cuticular hydrocarbon plasticity in three rice planthopper species’ describes the variation in cuticular hydrocarbons (CHCs) at different developmental stages, sex and environment along with transcriptomal changes leading to the CHC variation under various environment.
The manuscript is well written and described. The work has good future implications. However, there are few minor comments, which the authors might include/change in the manuscript.
Abstract
Line 17: lipid layer, which function…
Introduction
Line38. Avoid the damage caused by…
Line440. Remove blends
Methods
Line 517. Please include a brief description of the procedure used to extract CHCs. This will help readers have a quick understanding.
The authors should cite and mention the following papers in the text.
NlCYP4G76 and NlCYP4G115 Modulate Susceptibility to Desiccation and Insecticide Penetration Through Affecting Cuticular Hydrocarbon Biosynthesis in Nilaparvata lugens (Hemiptera: Delphacidae)
And, Advances in deciphering the genetic basis of insect cuticular hydrocarbon biosynthesis and variation.
